# On Representation of Natural Image Patches

## Abstract

To optimize survival, organisms need to accurately and efficiently relay new information throughout their systems for processing and responses. Furthermore, they benefit from predicting environmental occurrences, or in mathematical terms, understanding the probability distribution of their environment, based on both personal experiences and inherited evolutionary memory. These twin objectives of information transmission and learning environmental probabilistic distributions form the core of an organism's information processing system. While the early vision neuroscience field has primarily focused on the former, employing information theory as a guiding framework [3, 32, 19, 1, 9, 28], the latter is largely explored by the machine learning community via probabilistic generative models. However, the relationship between these two objectives has not been thoroughly investigated. In this paper, we study a biologically inspired information processing model and prove that these two objectives can be achieved independently. By evenly partitioning the input space to model input probability, our model bypasses the often intractable normalization factor computation. When applied to image patches, this model produces a sparse, nonlinear binary population code similar to early visual systems, with features like edge-detection and orientation-selective units. Our results not only offer potential new insights into the functioning of neurons in early vision systems, but also present a novel approach to represent natural image patches.

## 1 Introduction

Nature, through billions of years of evolution, has likely developed optimal methods for processing visual information within the constraints of biological feasibility. However, attempting to precisely emulate every detail of these biological systems [30, 21] in order to construct an optimal visual information processing model presents significant complexities, especially without a comprehensive understanding of the underlying principles.

In parallel, deep learning models, particularly Convolutional Neural Networks (CNNs), have demonstrated exceptional performance in various computer vision tasks, such as image classification, object detection, and semantic segmentation. Despite their success, these models still fall short of biological vision systems in several key areas such as the ability to generalize from limited data, robustness to variations, 3D understanding, and processing speed and efficiency.

Moreover, deep learning models pose several unresolved challenges. Firstly, their decision-making processes are often opaque, leading to the "black box" label. Secondly, deep learning architectures typically involve numerous layers without explicit functions associated with each layer [12, 8], unlike the biological brain where each stage of visual processing has a distinct role and purpose [7]. Lastly, he

sequence of information processing in deep learning models is largely dictated by their architectural structure. It is still a question how to ascertain when one processing stage is finished and when it's appropriate to pass information to the next layer.

Drawing inspiration from biological systems and prior studies [19, 1, 9, 28], and with a view to find an alternative approach to the current deep learning framework, this paper aims to explore the fundamentals of the first stage of an optimal visual information processing system. This exploration is undertaken incrementally, starting from a single pixel, and progressively advancing to image patches.

## 2 One Pixel

We begin with the simplest conceivable case, where the input to the model consists of a single pixel with one color channel. Although seemingly trivial, this model can represent various biological units. For example, it could model the eyespot of single-celled organisms like Euglena, the large monopolar cells found in an insect's compound eye or the bipolar cells in the retina. The single pixel case has been studied [14, 1]. We aim to review it to introduce the central concepts and the main theory, which will also be applicable to more complex scenarios.

Let us denote the light intensity of the pixel as $x$, and let $p(x)$ represent its probability distribution. We define an information processing unit (IPU) as a model that receives inputs and passes processed information to subsequent stages. As the first stage in an organism's information processing system, the single pixel IPU carries the same dual objectives as later stages: transmitting information and learning environmental probabilistic distributions. Therefore, the two objectives for the single pixel IPU are to transmit information about $x$ efficiently through its output and to learn $p(x)$.

Information is quantified by Shannon entropy. To compute the Shannon entropy of the input, we assume $x$ is a discrete variable with $M$ states, after all light intensity is quantized according to quantum mechanics, although $M$ could be a very large number, making $x$ practically indistinguishable from a continuous variable. The information obtained from the pixel when we know the intensity is $x$ can be represented as $I(x) = -\log p(x)$. The average information a state of the pixel contains, the Shannon entropy, is given by:

$$H_p = -\sum_{i=1}^{M} p(x_i) \log p(x_i). \tag{1}$$

The IPU transforms the input $x$ into the output $y = f(x)$, where the output space comprises $N$ distinct states. In contrast to previous studies that assumed one-to-one mapping between input and output, here we posit that $N \ll M$. This assumption is more congruent with biological constraints; for instance, the luminance resolution levels at synapse terminals in a zebrafish's retina are only about 10 [25]. Moreover, this assures that after processing the information is significantly reduced, thereby simplifying the tasks for subsequent stages.

The function $f(x)$ assigns $x$ into $N$ groups, each corresponding to a fixed $y$. We denote all $x$ values in group $j$ as $G_j$, and the size of this group as $n_j$. The entropy of the output is given by

$$H_Q = -\sum_{j=1}^{N} Q(y_j) \log Q(y_j), \tag{2}$$

where $Q(y)$ represents the probability distribution of the output states.

Previous research [14, 1] has primarily emphasized the first objective of an IPU, which is maximizing the rate of transmission [19]. This goal is particularly relevant for early-stage IPUs, where the distinction between signal and noise is not yet clear. Maximizing the rate of transmission is equivalent to maximizing $H_Q$ (see proof in Appendix A), leading to a constant $Q(y)$. Biological neurons have been observed to follow this coding scheme [14].

Simultaneously, an IPU should also strive to fullfil the second objective and model $p(x)$ as accurately as possible. Mathematically, this involves minimizing the Kullback–Leibler divergence between

$p(x)$ and the distribution learned by the IPU. This raises an interesting question: Are these two optimization objectives contradictory, or do they essentially represent the same task?

## 3 Even Code Principle

To determine how an IPU models $p(x)$ we need to translate the output probability distribution $Q(y)$ into the input space as $q(x)$. $q(x)$ is a step function:

$$q(x) = q_j, \text{ for } x \in G_j, \tag{3}$$

and we have the following relations:

$$Q(y_j) = \sum_{x \in G_j} p(x) = \sum_{x \in G_j} q(x) = n_j q_j. \tag{4}$$

Minimizing the difference between $p(x)$ and $q(x)$ can be achieved by minimizing their Kullback–Leibler divergence:

$$D_{KL}(p||q) \quad = \quad H_{pq} - H_p, \tag{5}$$

where $H_{pq}$ is the cross entropy. It can be proved that the cross entropy $H_{pq}$ is equal to the entropy of the learned distribution in the input space defined as (see proof in Appendix B):

$$H_q = -\sum_x q(x) \log q(x), \tag{6}$$

and we get

$$D_{KL}(p||q) = H_q - H_p. \tag{7}$$

Since $H_p$ is fixed, minimizing the KL divergence requires minimizing $H_q$. The previous question now transforms into understanding the relationship between maximizing the entropy of the distribution in the output space ($H_Q$) and minimizing the entropy of the learned distribution in the input space ($H_q$).

Suppose we have two adjacent zones in the transformed space where the corresponding $Q(y_1)$ and $Q(y_2)$ are not equal, let's assume $Q(y_1) > Q(y_2)$. One can reduce the inequality by shifting the boundary between these two zones and moving one x value from $G_1$ to $G_2$. This shift corresponds to a small change of probability, $\delta$, for both zones. Note that $\delta$ is comparable to $q_1$ and $q_2$, as we assume the distribution is smooth. We know that reducing the inequality of $Q(y_1)$ and $Q(y_2)$ always increases $H_Q$. If the two optimization problems are the same, then $H_q$ should increase; if they are contradictory, $H_q$ should decrease. The change of $H_q$ can be calculated as:

$$\Delta H_q = -[Q(y_1) - \delta] \log \frac{Q(y_1) - \delta}{n_1 - 1} - [Q(y_2) + \delta] \log \frac{Q(y_2) + \delta}{n_2 + 1}$$
$$+ Q(y_1) \log \frac{Q(y_1)}{n_1} + Q(y_2) \log \frac{Q(y_2)}{n_2} \tag{8}$$

$$= q_2 - q_1 + \delta(\log q_1 - \log q_2 + \frac{1}{n_1} + \frac{1}{n_2}) + O(\delta^2) + O(\frac{1}{n_1^2}) + O(\frac{1}{n_2^2}) \tag{9}$$

$$\approx q_2 - q_1 + \delta \log \frac{q_1}{q_2}. \tag{10}$$

The change can either be positive or negative depending on $q_1$ and $q_2$. Since minimizing $H_q$ and maximizing $H_Q$ are not contradictory, these objectives can be tackled independently. Given a fixed number of output levels $N$, we first maximize $H_Q$ to retain as much input information as possible. If further refinement of $p(x)$ modeling is required, we can increase the output resolution $N$.

The aforementioned reasoning extends naturally to multivariate scenarios, as no assumptions about one-dimensionality of the input were made. We articulate the goal of a general information processing unit as follows: An information processing unit (IPU) transforms input space with $M$ states into output space with $N$ states, where $N \ll M$. Given a smooth input probability distribution as $M \to \infty$ and a piecewise smooth transformation function, the *sole* goal of an IPU with a fixed output resolution $N$ is to yield an even output probability distribution, hence retaining maximum information from the input. To attain better modeling precision, the output resolution $N$ of the IPU should be increased. This will be referred to as the principle of even code. In the next sections, we will apply the even code principle to more complex inputs.

## 4  Two Pixels

For two pixels $(x_1, x_2)$, we can either use one IPU directly to model $p(x_1, x_2)$ or use two IPUs to model $p(x_1)$ and $p(x_2)$ separately, followed by another IPU to model the outputs $p(y_1, y_2)$. We will use the second approach, as processing as much information locally reduces the cost of information transfer. In fact, when images are stored on computers, gamma encoding is utilized to create an approximately even distribution of pixel values. When these images are displayed, pixel values undergo gamma correction to recover the original statistics for human eyes to process. In the following sections, we will assume that all pixel values $x$ have already been processed by dedicated IPUs, resulting in a roughly even probability distribution.

The probability distribution $p(x_1, x_2)$ of natural images is relatively simple. The majority of the probability is concentrated around the diagonal line $x_1 - x_2 = 0$, with $p(x_1, x_2)$ rapidly decaying as $|x_1 - x_2|$ increases (see Fig. 1 (a) for example). Intuitively, we can use lines parallel or/and perpendicular to $x_1 - x_2 = 0$ to divide the probability distribution into even partitions.

### 4.1  One Basis

To investigate how IPUs learn $p(x_1, x_2)$, we conduct numerical experiments using a multilayer perceptron (MLP) as the IPU to approximate $y = f(x)$ and model $p(x)$ [23]. Other function approximation methods may also be applicable. To partition the input probability distribution with one set of parallel lines, only one IPU with $N$ output nodes is needed. According to the even code principle, for each input, only one of the $N$ output nodes should be activated, and the probability of activating any one of the $N$ output nodes should be equal. We use the softmax function as the last layer of the MLP to ensure each output value is within [0, 1], and that if a node is activated (output value equals 1), it is the only node being activated. We use stochastic gradient descent and the following loss function to train the MLP:

$$E = \sum_i \langle y_{si} \rangle_s \log \langle y_{si} \rangle_s + k \langle - \sum_i y_{si} \log y_{si} \rangle_s. \tag{11}$$

$y_{si}$ represents the value of the i-th output node for the s-th input sample, while $\langle \rangle_s$ denotes the average over all samples in a training batch. The first term in the loss function ensures each output node has an equal chance to be activated on average. The second term promotes activation of only one node per input while suppressing the remaining nodes, mimicking lateral inhibition when combined with the softmax function. The factor $k$ balances the two terms to achieve the desired result. Fig. 1 (a) show the results learned by MLPs with 16 output nodes.

### 4.2  Multiple Bases

To partition the input space with two sets of orthogonal lines we need two MLPs. The orthogonality is achieved by enforcing

$$Q(y_1, y_2) = \frac{1}{N_1 N_2}, \tag{12}$$

where $N_1$ and $N_2$ represent the number of output nodes of the two MLPs (refer Appendix C for proof). If more than two orthogonal bases are required for partitioning the space, we can enforce Eq. (12) for each combination of two bases to ensure orthogonality between them. The loss function for multiple orthogonal bases with independent states is

$$E = \frac{1}{\binom{B}{2}} \sum_{<b,b'>} \sum_{ij} \langle y_{bsi} y_{b'sj} \rangle_s \log \langle y_{bsi} y_{b'sj} \rangle_s + \frac{k}{B} \langle - \sum_{b=1}^{B} \sum_i y_{bsi} \log y_{bsi} \rangle_s, \tag{13}$$

where $b$ is the base index, and $B$ is the number of bases. $\sum_{<b,b'>}$ denotes the sum over all $\binom{B}{2}$ combinations of two distinct bases. $y_{bsi} y_{b'sj}$ is the probability $Q(y_b, y_{b'})$ for the sample s when $y_b$ and $y_{b'}$ take their i-th and j-th value respectively.

Fig. 1 (b) shows an example of two-pixel input space partitioning using two orthogonal bases. For a more detailed discussion on the experiments and additional results, refer to Appendix D.

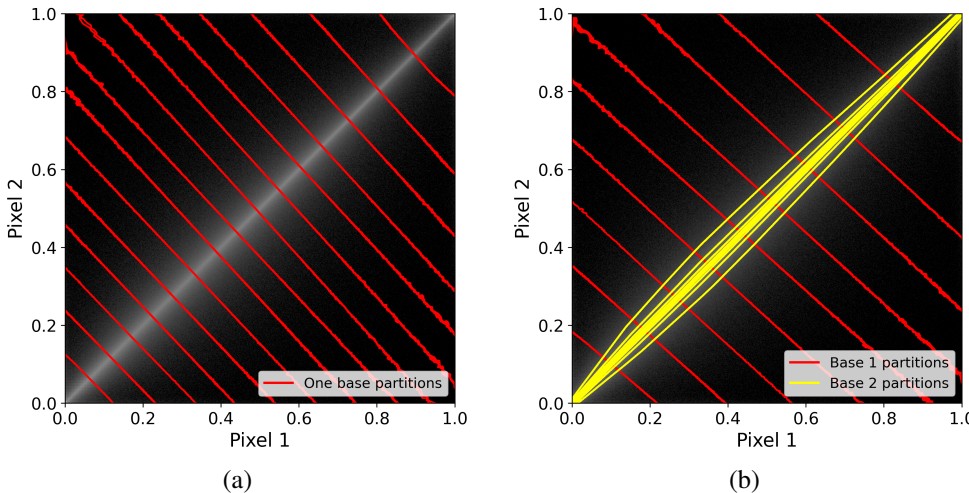

Figure 1: Evenly partitioning the two-pixel probability distribution learned by multilayer perceptrons (MLPs). The X and Y axes represent the rescaled intensities $x_1$ and $x_2$ of the two pixels in the range [0, 1]. The quantity $n(x_1, x_2) + 1$ is plotted in gray on a log scale, where $n(x_1, x_2)$ denotes the number of occurrences of the two-pixel values among the sampled data. Color lines indicate the boundaries of states for each basis learned by an MLP, with one color representing one basis. (a) One basis with 16 independent states, which partitions the space based on the total intensity $x_1 + x_2$. (b) Two orthogonal bases, each with 10 independent states, dividing the space based on the total intensity $x_1 + x_2$ and the contrast $x_1 - x_2$ approximately.

Additionally, it's worth noting that orthogonal bases with independent states might model grid cells [11] in the entorhinal cortex, though this topic is beyond the scope of the current paper.

# 5   Image Patches

Next we move on to study gray and color image patches. We use **x** to represent the vector of input pixel values of the image patch. The multivariate input probability distribution $p(\mathbf{x})$ is considerably more complex compared to the previous examples. If we use only one basis to discretize the input probability space (e.g. Fig. 1 (a) in Appendix D), the required number of independent states for a good approximation would be very large, making the evaluation of the softmax function computationally expensive. On the other hand, using multiple orthogonal bases would also significantly increase the computational cost to ensure orthogonality if the number of bases is more than just a few. Additionally, determining the optimal number of bases and the number of independent states for each basis are challenging.

Aside from the computational cost, another issue arises when working with image patches: we want the representation to capture the similarity between inputs. However, using orthogonal bases with independent states makes it difficult to gauge input similarity through methods such as calculating the difference between representations, even if we can establish an order for the states of each basis. Therefore, we need a more suitable coding scheme for complex inputs like image patches.

Real-valued vectors are a natural choice, given their extensive use in representing a variety of entities such as images, texts, and categorical variables [13, 22, 10]. The norm of the difference between two vectors can function as a measure of similarity. Nevertheless, if we want the representation **y** to mirror input similarity, each value of **y** should encapsulate all samples perceived as identical within the same group $G$. Under this constraint, $Q(\mathbf{y})$ cannot remain constant, thereby conflicting with the even code principle.

The resolution to this conflict involves permitting the representation to mirror input similarity at the most granular level, while enforcing the even code principle at a larger scale in the transformed space. We will detail this method in subsequent sections.

## 5.1 Loss Function

To promote even distribution, we incorporate a loss function that compels input samples to repel each other in the transformed space. This repulsive force diminishes with increasing distance, as described by the following equation:

$$E = \langle -\ln |\mathbf{y}_s - \mathbf{y}_{s'}| \rangle_{<s,s'>}. \tag{14}$$

Here, $-\ln |\mathbf{y}_s - \mathbf{y}_{s'}|$ represents the potential energy due to the repulsive force, which is proportional to the inverse of the Minkowski distance between the representations of samples $s$ and $s'$ in the transformed space. Alternative forms of potential energy and distance measures could also be applicable. $\langle \rangle_{<s,s'>}$ denotes the average over all sample pairs.

Should numerous samples converge at one point in the transformed space, they will exert a strong repulsive force in the surrounding area, thereby discouraging other samples from occupying nearby positions. To prevent samples from pushing each other infinitely far apart, we restrict the representation values to be within the range $[0, 1]$. With this constraint, the repulsive force pushes samples towards the vertices of the unit hypercube, effectively reducing the representations from real vectors to binary vectors. As a result, an even distribution is achieved on a larger scale in the transformed space, which consists solely of the vertices.

In the context of binary vectors, the collection of output nodes can be viewed as a vocabulary, and activated nodes by an input image patch act as its representative tokens. Unlike fixed-length representations with real-valued vectors, binary representations can employ fewer tokens for more common image patches (e.g., homogeneous patches), and more tokens for less common, structurally-rich patches. This can be accomplished by introducing a second term, $\langle |\mathbf{y}_s| \rangle_s$, to the loss function, which echoes the sparsity regularization term found in various studies [9, 16, 26, 27, 29, 4]. The updated loss function becames:

$$E = \langle -\ln |\mathbf{y}_s - \mathbf{y}_{s'}| \rangle_{<s,s'>} + \alpha \langle |\mathbf{y}_s| \rangle_s, \tag{15}$$

where $\alpha$ is a free parameter to adjust sparsity.

In practice, we add a small value $\epsilon = 10^{-38}$ to the distance, allowing slightly different samples to share the same representation and enhancing numerical stability. Another approach to improve numerical stability involves using a theoretically equivalent form of the loss function, which instead of allowing samples to repel each other in the output space, we enable nodes to repel one another, encouraging output nodes to be as independent as possible [24].

## 5.2 Experiments

In the following experiments, we use either a single MLP with N outputs, or N MLPs each with one output, as the IPU to approximate the transformation function $\mathbf{y} = f(\mathbf{x})$ and model $p(\mathbf{x})$. The last layer of the MLP is a sigmoid layer, ensuring the output value ranges between 0 and 1. Our training data comprises random image patches extracted from the COCO 2017 image dataset [18] or the ImageNet dataset [6]. No image preprocessing is used. Additional training details are provided in Appendix E.

### 5.2.1 Output Statistics

First, we examine the statistics of the learned representation. Across all experiments, we observe qualitatively similar output statistics, irrespective of the IPU architectures and training specifics, provided the training has properly converged. For illustration, we present an example using a model trained on $5 \times 5$ color image patches. It uses 96 MLPs, each with one output node and a middle layer of 48 nodes, as the IPU. Following training, the model is used to generate representations for 1 million random image patches for this analysis.

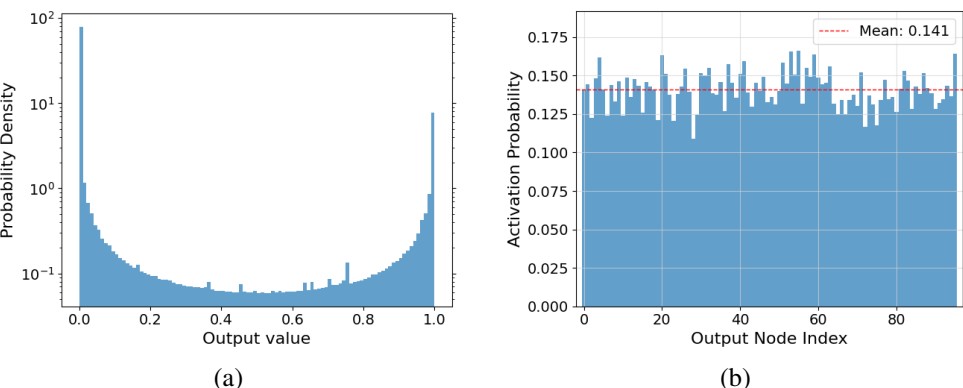

Figure 2: Statistical analysis of the learned representation using the loss function Eq. (15). (a) Histogram of the model's output values on a log scale. (b) Probability of an output node being activated by a random image patch.

Fig. 2 (a) presents the histogram of output values on a logarithmic scale. The vast majority of output values are either at 0 or 1. As such during inference, we can round the outputs to yield a binary representation. Fig. 2 (b) illustrates the probability of an output node being activated by a random image patch. All nodes demonstrate similar activation probabilities, indicating an even distribution at this coarse scale. Further statistical analysis of the output representation is available in Appendix F.

### 5.2.2 Image Patch Similarity

Next, to examine how the learned representation reflects the similarity between image patches, we display 16 random image patches, each followed by 9 image patches similar to them in the binary representation space, as shown in Fig. 3 (a). The same $5 \times 5$ color image patch model is used. The learned representation clearly captures perceptual similarity. The results shown in Fig. 2, Fig. 3 (a), and additional results in Appendix E confirm that with the loss function Eq. (15), we can indeed learn a sparse binary representation which reflects the image similarity while adhering to the even code principle.

For comparison with a traditional convolutional neural network, we present the results generated with the first 10 layers of a VGG16 model [31] pre-trained on ImageNet in Fig. 3 (b). The image patch representation from the first 10 layers of the VGG16 model is a float vector of size 128. The even code model, with only 96 binary outputs, achieves results similar to the VGG16 model. These 96 binary outputs occupy the same storage space as a float vector of length 3 — just 1/42 of the VGG16 representation's size, which underscores the exceptional efficiency of the even code method in image patch representation.

### 5.2.3 Local Edge Detectors and Orientation-Selective Units

Biological visual systems' initial stages are known to possess local edge detectors and orientation-selective units [17, 2]. While CNNs have been successfully trained to detect boundaries via supervised learning [20], their initial layers have not shown proficiency in edge detection [15]. Notably, prevalent local edge detection algorithms, such as the Canny edge detector [5], still primarily rely on non-deep learning methods.

Does the even code model, proposed as the initial stage of an optimal image processing system, resemble biological systems more closely? To answer this, we trained an even code model on $4 \times 4$ grayscale image patches and applied it to images with a stride of 1 pixel, generating feature maps for each output node. The model comprises a MLP with 64 outputs and an intermediary layer with 100 nodes. Fig. 4 illustrates the feature maps of 4 output nodes of the even code model for 4 different input images. It also shows edges generated by the Canny edge detector as comparison. Interestingly,

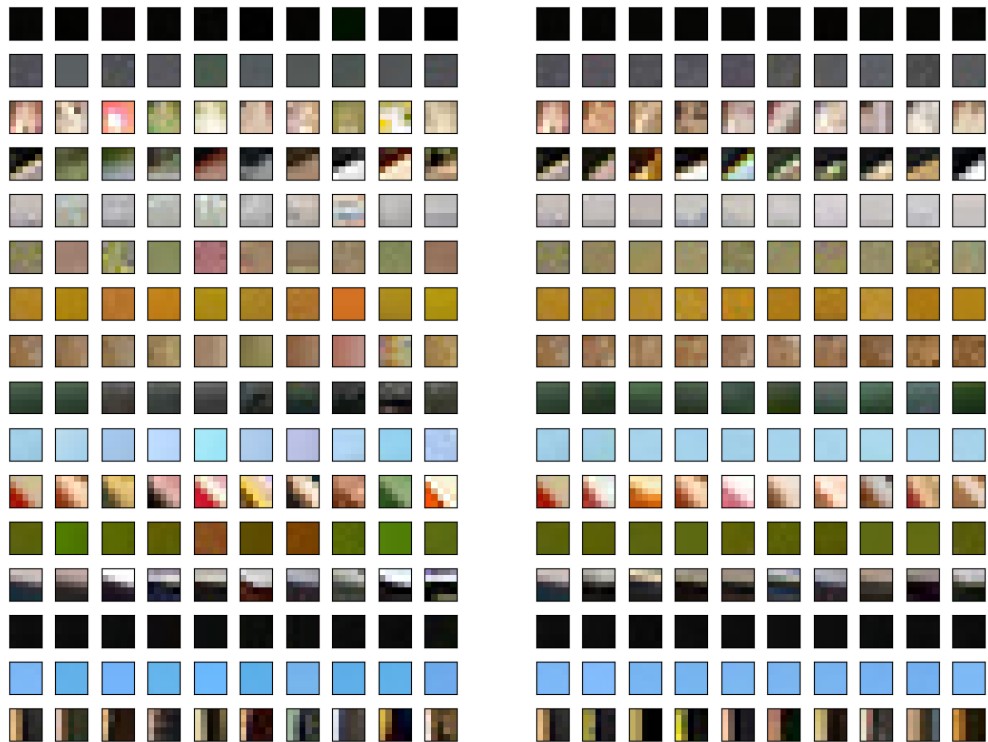

(a) Even code model with 96 binary outputs      (b) Early layers of VGG16 with 128 real outputs

Figure 3: Image patches with the shortest distance in the representation space to 16 randomly selected image patches. The first column presents the 16 random image patches, while the succeeding nine columns display patches that are closest to the first-column patches in the same row. (a) Distances are computed using using an even code model with 96 binary outputs. (b) Distances are computed using the first 10 layers of a pretrained VGG16 model with 128 real outputs.

with this simple network architecture, the even code model demonstrated a remarkable capability in edge detection, rivaling the multi-stage Canny edge detector.

Furthermore, Fig. 5 shows the feature maps of 5 output nodes for a sample bike image. Spokes of different orientations activate different nodes, indicating that these output nodes of the even code model have varying orientation preferences, similar to orientation-selective units found in bilogical systems.

# 6  Conclusion

In summary, this paper demonstrates that maximizing the information-carrying capacity of output channels and modeling the input probability distribution are not mutually exclusive objectives and can be pursued independently. Given a specific output resolution, the sole goal of an information processing unit is to preserve as much information from the input as possible by ensuring an even distribution of samples in the output space. We applied the even code principle to study the probability distributions of two-pixel systems and image patches. For the two-pixel system, we learned orthogonal bases with independent states to model its probability distribution. For image patches, the even code approach naturally led to a nonlinear sparse binary representation. The even code model also shares additional similarities with early visual systems, such as the presence of local edge-detecting and orientation-selective units. These similarities suggest that the even code model could potentially serve as a new representation for neurons in early visual systems.

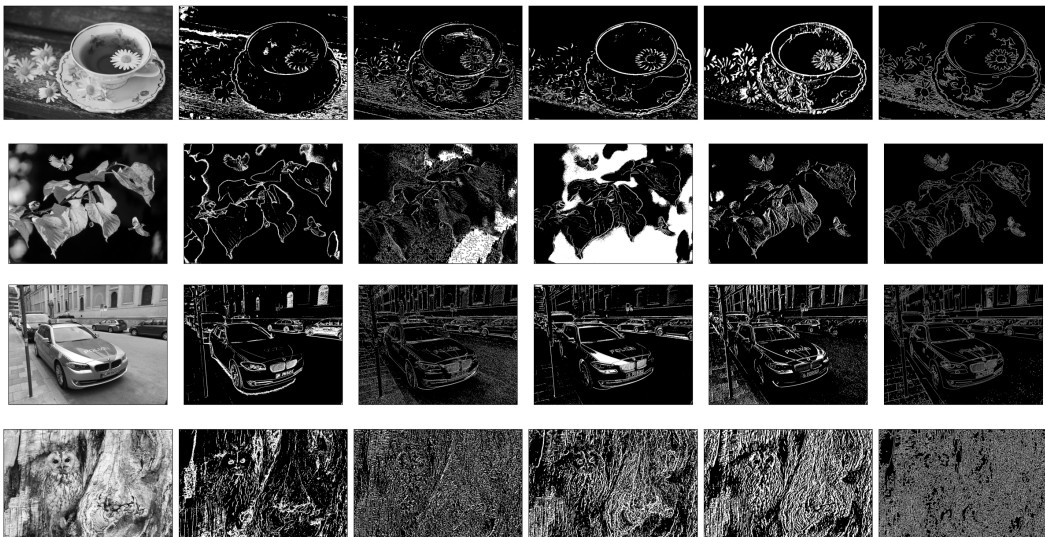

Figure 4: Feature maps of nodes resembling local edge detectors. The first column presents four grayscale test images. Each subsequent column, except the last one, displays the feature maps corresponding to the same output node for the four test images. The last column shows edges generated by the multi-stage Canny edge detector for comparison.

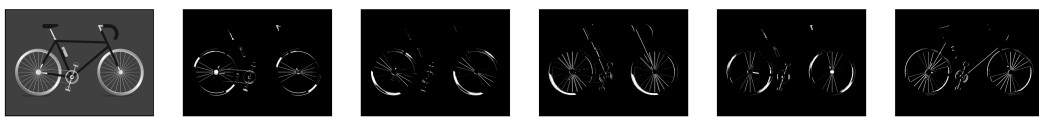

Figure 5: Feature maps of five orientation-selective nodes applied to a test image. Spokes in different orientations activate distinct nodes, illustrating the orientation-selectivity of these nodes.

There are several intriguing directions for future research. First, the even code model has been applied to inputs ranging from as simple as one pixel to more complex color image patches. Can we extend its application beyond the early stage of visual information processing? Second, the even code model could be extended to videos by incorporating an additional time dimension alongside color, width, and height dimensions. Investigating time-varying inputs, which produce spike train-like outputs, and conducting an in-depth comparison with early visual systems would be very interesting. Third, the even code model can also be extended to binocular vision data by adding another input dimension of size two. Whether the model with binocular and/or video data can construct a 3D model of the world based on data of two spatial dimensions is an intriguing question. Fourth, while this paper focuses on visual information, the even code model is a general method that could be applied to model other multivariate probability distributions as well. Lastly, on the application side, the even code model has potential in various areas, including local edge detection, image and video compression/denoising/retrieval, texture classification, and multispectral/hyperspectral image processing.

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
