# OpenReview forum: "On Representation of Natural Image Patches"
_NeurIPS.cc/2023/Conference — Submitted to NeurIPS 2023_

### Official Review · Reviewer_1uCk · 2023-07-03

**Soundness:** 3 good
**Presentation:** 3 good
**Contribution:** 3 good
**Rating:** 3
**Confidence:** 3

**Summary:**

- This study proposes a new framework to combine neural coding concepts of information transmission and probability density modeling.
- This framework is based on an even code principle where the output response density strives to be even, given some arbitrary input density.
- The authors show that this coding principle produces sensible bases for low-dimensional inputs, and orientation-tuned filters for natural image patches.
- While conceptually straightforward, it is unclear to me whether this study provides unique insight into sensory coding in neural populations.

UPDATE: Sep 1, 2023. I have read the rebuttal, and maintain my score (see details below).

**Strengths:**

- The study is clearly presented.
- The concepts of max/min entropy on the output and input densities are conceptually simple to follow.
- The numerical experiments are reasonable.

**Weaknesses:**

- This paper begins with what seems like a false dichotomy of information transmission vs. sensory probability density modeling. Indeed, from a pragmatic point of view, how can one guarantee optimized information transmission without having a good density model of the signal to be transmitted? There exists literature in this area (see questions section), and the motivation/framing of this present study is concerning.
- There is a bit of a conceptual leap from 1 or 2 pixels to full image patches, with additional complexity and machinery introduced. The described rationale seems reasonable enough, but it is unclear whether the two-pixel orthogonal case can provide adequate intuition for the multi-dim case. Would a 2D non-orthogonal example be illuminating at all?
- Unclear to me whether these results, which rely on binary coding provides theoretical insight for real neural coding. Spikes are inherently binary, yes, but typically spike counts/rate are what is considered the informative variable in neural coding.

**Questions:**

- I'm surprised there's no mention of the work by Ganguli and Simoncelli ("Efficient sensory encoding and Bayesian inference with heterogeneous neural populations", Neural Comp. 2014), which directly integrates density modeling with information transmission in neural populations. These authors showed that synaptic weighting functions and neural population responses are arrange so as to implicitly encode the probability of the sensory signal. There have also been attempts at generalizing these concepts to multi-dimensional stimuli by Yerxa et al ("Efficient sensory coding of multidimensional stimuli"; PLOS Comp Biol. 2020). At the very least I believe the present study should discuss how their approach fits in with this existing literature.
- Minor point: is "even code" a standard information theory term? If it is then that's fine, but it seems like equalized probability would be a more descriptive term.

**Limitations:**

There was no discussion of limitations. Unclear to me what the drawbacks are of this approach compared to existing literature.

---

> ### Author Rebuttal · Authors · 2023-08-03
>
> We sincerely appreciate the time and effort you have spent in evaluating our paper.
>
> * Regarding the concern on a false dichotomy
>
> In this paper, we aim to explore what an optimal early-stage information processing system would look like from first principles with as few assumptions as possible. We have identified the two fundamental goals. To start, it is crucial to examine the relationship between them carefully. We have indeed found that these two goals are not the same, and there's no guarantee that optimizing one goal will simultaneously optimize the other. As far as we know, this divergence has not been addressed before.
>
> * Regarding the conceptual leap
>
> We have tried to apply the method used for two pixels directly to image patches. It works, and the results are reasonable. However, it has a few problems, as we mentioned in the paper. We were stuck here for more than a year until we developed the current method described in the paper. In essence, the methods follow the same principle: the loss functions attempt to make the responses of neurons as different/orthogonal/uncorrelated as possible, either by enforcing output statistics explicitly as in the case for two-pixel systems, or by achieving this implicitly by allowing the response vectors to repel each other as in the image patches cases. The method for image patches is close to the two-pixel orthogonal case, where each base has only two states. It essentially does the following:
>
> a) We allow each neuron to react to a sequence of image patches and get a response vector for each neuron.
>
> b) We use a loss function to make the response vector for each neuron as unique (less correlated) as possible.
>
> * Regarding "2D non-orthogonal example"
>
> Perhaps you are referring to something like Fig. 1 in the attached pdf in global rebuttal? In this figure, we created an artificial 2D normal distribution and used one base to model it. Hexagonal lattice is generated by the model itself. It resembles some results in Yerxa's paper (our states mapped to their neuron).
>
> * Regarding the two papers by Ganguli et al. and Yerxa et al.
>
> They are indeed very relevant. Had we not overlooked these papers, we could have formulated the abstract and introduction more precisely and effectively. Here is the comparison:
>
> 1. Essentially, the output states in our model correspond to neurons in their model. Using each state to represent an equal portion of $p(x)$ implies that the density of states is proportional to $p(x)$ in the continuous limit. This agrees with their result.
>
> 2. Their analysis assumes the density of neurons is continuous (e.g. above Eq. 2.10), i.e. implicitly assumes an infinite number of neurons, while we assume a finite number, which is more realistic. With infinite states, regardless of how these states are arranged, $p(x)$ is always perfectly approximated. This is likely why $p(x)$ modeling is not seen as a problem by them. The problem will arise in a more realistic setting.
>
> 3. They use many assumptions and manually select features, such as rate coding, specific form of neuron tuning curves or a hexagonal lattice in the 2D case. While these assumptions are resonable or widely used they impose limitations. In our ab initio approach, we aim to capture the essence of an information processing system with only three main assumptions:
>
>     a) An information processing system needs to achieve two fundamental goals.
>
>     b) For an IPU, the number of output states N is significantly smaller than the number of input states M.
>
>     c) In the limit M → ∞, the input is continuous, and the transformation is a (piecewise) smooth function.
>
>     Assumption a) is nearly axiomatic. Assumption c) is also implicitly used by Ganguli's work and many allowing for the calculation of derivatives. Therefore only assumption b) is new.
>
> 4. For the model used by the two papers and many others, the mutual information between inputs and outputs, to cite them:
>     > is notoriously difficult to compute (or maximize) as it requires summation or integration over the high-dimensional joint probability distribution of all possible stimuli and population responses.
>
>     They instead chose to optimize a lower bound on mutual information.
>     In our model, the mutual information equals to the output entropy, which is not only much easier to deal with analytically and numerically but also potentially more straightforward for the neural system to implement as it only requires local information at the output.
>
> 5. Our theory has been numerically applied above 2D and can also be readily extended to time-varying input.
>
> * Regarding whether the results provide insight for real neural coding
>
> With spike count conveying information, it still implies that the output has a finite resolution. If each neuron has N output states and receives input from L neurons, then the number of input states is $N^L$, which is much larger than $N$. This is in agreement with assumption b), thus making our theory applicable. For rate coding, the methods used in the two-pixel models may be more relevant. The method developed for image patches may be more related to temporal coding. Our theory also aligns with certain experiments, such as *Decorrelated Neuronal Firing in Cortical Microcircuits* by Ecker et al.
>
> * Regarding the limitation of the paper.
>
> We acknowledge the following limitations compared with others:
> 1. Our study is abstract and does not emulate some key aspects of real neurons, such as spiking activity. This makes comparison with existing neuroscience literature not straightforward.
> 2. This paper only studies the noiseless case.
>
> * Regarding the name
>
> It is not a standard information theory term. We wanted a short name to refer to our method and came up with this term, which means evenly distributed probability code or evenly partitioned probability code. We are considering more descriptive terms.

---

> > ### Comment · Reviewer_1uCk · 2023-08-11
> >
> > Thanks and I appreciate your earnest responses to my comments and questions.
> >
> > I do believe that you have a novel concept/idea in this study that may be worth presenting. But, in its current form, there is lack of proper contextualization with respect to highly relevant work; and, it's a bit too abstract to understand how one might try to relate this back to neural coding in biological systems (e.g. testable predictions, fits, etc.). I believe remedying these pitfalls would undoubtedly drastically affect much of the manuscript, including the overall narrative of the study.
> >
> > For these reasons, I'm sorry to say that I am not inclined to change my score.

---

### Official Review · Reviewer_KcKg · 2023-07-05

**Soundness:** 3 good
**Presentation:** 3 good
**Contribution:** 2 fair
**Rating:** 4
**Confidence:** 4

**Summary:**

This paper presents a method for the representation of elementary natural images, based on the observation that classical studies in computational neuroscience focus mainly on methods to improve code efficiency, but that this could be complemented by a study of probability density modeling between neighboring pixels to improve image representation. This work consists in studying a coding principle based on a probabilistic representation and its formalization in a form of variational optimization. The paper presents the elementary method for a single pixel, then extends it to two pixels, and applies it to  small images extracted from natural images. This method is enhanced by a heuristic that allows  to formulate a cost function and thus derive an optimization algorithm. The results allow  to numerically validate this principle by deducing output statistics, as well as the emergence of local contour detectors.

**Strengths:**

A major strength of this paper is that it derives the image representation algorithm from fundamental principles of machine learning, particularly probabilistic representations. In this way, it rigorously defines the problem of establishing dependencies between the luminance values of neighboring pixels.


**Weaknesses:**

The first limitation of this paper is that it applies to very elementary signals, i.e. a pixel, a pair of pixels, or small images of dots. As the initial aim of the paper is to understand the computational functioning of the biological networks that underlie the efficiency of vision, this approach is extremely caricatural, and dismisses many fundamental aspects, such as the largely parallel processing of large images, the use of large neural networks, or the ability to process multimodal images, in color or in motion, or more generally hierarchical processing that can be forward, but also modulated by feedback signals. Finally, the results that have been obtained, for example for the detection of local elementary contours, are difficult to interpret quantitatively and seem very preliminary.

**Questions:**

I have a number of questions arising from reading the paper.

First of all, it seems that the principle of optimization, and in particular its derivation for elementary changes in probability, is widely known in the literature. Can you highlight the originality of your approach compared with other probabilistic optimization methods, in particular all variational optimization problems? Figure 1 shows a dependency between pixel values. What is the relationship between your study and studies that have been carried out for many years, for example by Eero Simoncelli on divisive normalization? Also, the result shown in figure 2 shows an adaptation of the code according to the probability density. Are these results compatible with the homeostasis phenomena highlighted in biological neural systems? Finally, figure 5 shows the emergence of a representation of contours in an image, reminiscent of that obtained in studies of sparse coding, for example via Bruno Olshausen's framework. What is the relationship between your principle and these algorithms?

**Limitations:**

Finally, these questions about the paper reveal the main limitations of this work.

In particular, the introduction to the paper presents at length principles that seem very general, such as Shannon entropy, and the rest of the paper does not sufficiently highlight the novelties that are brought forward. This brings to light a main limitation of the paper, which is the fact that the propositions that are put forward are very ambitious, but the results are applied to very limited situations.

---

> ### Author Rebuttal · Authors · 2023-08-04
>
> Thank you for your valuable review. We will address your concerns point by point.
>
> * Regarding the limitation to elementary signals
>
> Our aim is to explore what an optimal early-stage information processing system would look like, from first principles with as few assumptions as possible. We assume that biological systems should be close to optimal, and we wish to compare which aspects of our results align with biological systems. We believe that this approach can shed some light on how early-stage biological systems function. We acknowledge that our approach is simplistic, but it's foundational to examining more intricate aspects of the visual system later on. We will enhance the abstract and introduction to clarify this goal.
>
> * Regarding the lack of quantitatively interpretable results
>
> We acknowledge this weakness. Our work, an abstract study from first principles, doesn't fully emulate real neurons, complicating comparisons with existing neuroscience literature. However, it can align with certain experiments, such as *Decorrelated Neuronal Firing in Cortical Microcircuits* by Ecker et al.
> On the other hand, the work is a theoretical exploration and not intended to solve a practical problem. This makes it difficult to compete with state-of-the-art methods in computer vision, at least initially (We did have one instance in Fig. 3).
>
> * Regarding the relation with divisive normalization
>
> Both methods aim to achieve statistical independence and both methods use non-linear local transformations. However, our method does not assume a specific form of transformation and thus is more general. Additionally, we are employing a completely different approach to learning the transformations.
>
> * Regarding the relation with homeostasis
>
> For biological neural systems to implement the even code principle, it is highly probable that they employ mechanisms such as homeostatic plasticity as well as lateral inhibition. This is indeed a very intriguing direction for future research.
>
> * Regarding our approach compared with probabilistic optimization methods and the originality of the derivations
>
> Variational optimization methods like variational autoencoders and other variational Bayesian techniques also try to approximate probabilities. However, their objectives and methodologies substantially differ from ours. Firstly, they focus on the posterior distribution, while we are only interested in the probability distribution of the input. Secondly, KL-divergence (or ELBO) is part of their loss function, while our loss function does not include this term.
>
> The reason is that we aim not only to learn $p(x)$ but also to optimize information transmission. In section 3, we have proven that these two goals are not identical. Consequently, one cannot typically find a solution where both goals are optimally achieved.
>
> Therefore, we propose a step-by-step approach to achieve these two goals. Firstly, we optimize information transmission, which also approximates $p(x)$ using a step function. We obtain the optimal solution for transmitting input information, but it is not the optimal solution for approximating $p(x)$ when the resolution is fixed. If the approximation of $p(x)$ is insufficient, N is increased to achieve this goal. The rationale behind this is that for living organisms, optimizing information transfer is a more urgent and crucial task than precisely learning $p(x)$.
>
> So, the derivations are necessary to understand the relationship between the two goals, and the method is new. We also have not observed any prior work on learning an image patch representation solely through an unsupervised method using a loss function, which only considers the outputs, not to mention the efficiency exceeding that of deep learning methods by a considerable margin. Contrastive learning, such as the one used in *A Simple Framework for Contrastive Learning of Visual Representations*, may seem like a counterexample, but it still requires labels and, therefore, is not a purely unsupervised learning method.
>
> (Note: On line 89, we state that "minimizing the KL divergence requires minimizing $H_q$". This should not be confused with the well-known proof that minimizing the KL divergence is equivalent to performing maximum likelihood estimation.)
>
>
> * Regarding how our approach compares with Olshausen's
>
> The current work is inspired by Olshausen's and many other related works. Here's what's new in our paper:
>
> 1. Many optimization methods, including those by Olshausen, use image reconstruction error minimization as one optimization goal. However, this is merely a reasonable first approximation. As stated in "Synaptic energy efficiency in retinal processing" (Vincent and Baddeley):
> > If the signal (the images) and the noise are Gaussian, then minimising mean squared reconstruction error maximises the information that the outputs provide about the inputs (Baldi & Hornik, 1995). It is known that natural images are not Gaussian distributed, but we would propose this as a reasonable first approximation.
>
>    Our method directly maximizes the rate of transmission without any approximation.
>
> 1. To calculate image reconstruction error, one needs to know the value of both input and output, and one needs to calculate the input from the outputs. Given these requirement, such a method may lack biological plausibility. Conversely, our method, which solely requires local knowledge of the output, offers a more biologically plausible model for neural implementation.
>
> 2. Our methods are guaranteed to generate near-optimal utilization of output channels thus are very efficient. The representations for image patches learned by our simple model are even comparable to sophisticated deep learning methods while only using less than 3% of the deep learning method's storage space. (Fig. 3)
>
> 3. Olshausen's method learns linear filters while our method is non-linear.
>
> 4. Our method can be easily extended to study time-varying inputs like videos.

---

### Official Review · Reviewer_Q2NW · 2023-07-07

**Soundness:** 3 good
**Presentation:** 3 good
**Contribution:** 2 fair
**Rating:** 3
**Confidence:** 4

**Summary:**

This paper explores the relationship between the information theory approach and the probabilistic generative model approach in the context of understanding neural coding. The author suggests that maximizing the information-carrying capacity of output channels and modeling the input probability distribution can be pursued as independent dual objectives. To investigate this hypothesis, the author begins by examining a one-pixel system, followed by a two-pixel system, gradually progressing to 2D image patches. The resulting codes obtained for the images exhibit similarities to edge detectors and orientation-selective neurons in V1, akin to many efficient coding models developed over the past two decades.

**Strengths:**

The presentation is reasonably clear.  It is rather interesting that the author begins by examining a one-pixel system, followed by a two-pixel system, gradually progressing to 2D image patches.

**Weaknesses:**

While the idea that both information transmission and probabilistic modeling of the images should be taken into consideration simultaneously might be new,  and is sufficient to learn edge detectors and orientation-selective neurons, the author has not established it is a necessary condition. In fact, literature in the last thirty years (from Law and Cooper's to Olshausen and Field and many others)  that such codes can be learned based on either one of the criteria.

It is surprising that the V1 neural codes were assumed to be sparse binary codes. What is the evidence?  The distribution of output values as shown in Figure 2a has not been observed biologically.  This brings the Even Code hypothesis into serious question.


**Questions:**

What is the evidence for  sparse binary codes?  The distribution of output values as shown in Figure 2a has not been observed biologically.  This brings the Even Code hypothesis into serious question.

**Limitations:**

Societal impact not discussed.

---

> ### Author Rebuttal · Authors · 2023-08-04
>
> Thank you for your review and insightful feedback. We appreciate your time and effort in providing us with valuable comments. We will address your concerns and questions in detail.
>
> * Regarding the learning of edge detectors and orientation-selective neurons in many previous studies
>
> The aim of this paper is not to provide yet another theoretical explanation for the emergence of edge detectors and orientation-selective neurons. Our goal is to explore what an optimal early-stage information processing system would look like, starting from first principles with as few assumptions as possible. We assume that biological systems should be close to optimal, and we want to compare which aspects of our results are in common with biological systems. We hope that with this approach, we can shed some light on how early-stage biological systems operate from a new perspective. We do find that in our solution there exist edge detectors and orientation-selective nodes, but this is more like a beneficial byproduct.
>
> What sets our work apart from many studies over the past three decades are:
>
> 1. Guaranteed by a rigorous first-principle method, our representation is optimal with independent output nodes. The efficiency of the representation surpasses that of the state-of-the-art deep learning models by a significant margin (Fig. 3).
> 2. The optimal representation can be learned by requiring only local knowledge at the outputs. This offers a more biologically plausible model for neural implementation compared to many works from the past three decades, including Olshausen and Field's, which requires non-local information of both input and output.
> 3. In our theory, we do not assume rate coding or temporal coding, thus we are coding-agnostic.
>
> * Regarding the use of "sparse binary codes" to describe neural coding in the early visual system
>
> We acknowledge that "sparse binary codes" is not a scientifically precise term. We intended to refer to "sparse binary signals". The exact coding scheme of the neural system is still not fully understood. While sparse coding is more generally accepted, neurons are often thought to use firing rates to convey information, which is not a binary code.
>
> Additionally, it's worth noting that our model is compatible with rate coding for the following reason:
>
> If a real neuron employs rate coding and uses spike count to convey information, it implies that the output possesses a finite resolution. If each neuron has N output states and receives input from L neurons, then the number of input states is $N^L$, which is significantly larger than N. This is in line with our definition of an Information Processing Unit (IPU). For rate coding, the methods used in the two-pixel case are likely more relevant (while we used the two-pixel case as an example, the method can be applied in higher dimensions). The method developed for image patches may be more related to temporal coding. However, careful studies are necessary and might require taking noise and time-varying inputs into account. These are interesting directions for future work.
>
> * Regarding Figure 2a
>
> The current study is abstract, and the model we propose does not try to model some specific aspects of real neurons, such as spiking activity. Therefore, Fig 2a cannot be directly compared to real neurons. The true meaning of Fig. 2a is as follows: In our numerical experiments, the initial setup is such that outputs can be any real value between 0 and 1. After the model has been trained, we have verified in Fig. 2a that almost all of the output values are either 1 or 0, signifying that our model encoded the images using binary representation.
>
> * Regarding biological evidence
>
> Our work, an abstract study from first principles, doesn't fully emulate real neurons, making comparisons with existing neuroscience literature more complex. However, it can align with certain experiments, such as *Decorrelated Neuronal Firing in Cortical Microcircuits* by Ecker et al. In Ecker's paper, they found:
>
> > We found that even nearby neurons with similar orientation tuning show virtually no correlated variability.
>
> This finding is in line with the even code principle. In fact, one version of the loss function in our image patches method aims to make the output nodes as decorrelated as possible.

---

> > ### Comment · Reviewer_Q2NW · 2023-08-20
> > **Thank you for your clarifications**
> >
> > I have read the reply by the authors. I agreed that there are some interesting and novel ideas there, but probably a bit too abstract, and need more contextualization, explanation and connections to biology to make the theory more concrete in order to have real impact.

---

### Official Review · Reviewer_aZBn · 2023-07-14

**Soundness:** 2 fair
**Presentation:** 1 poor
**Contribution:** 2 fair
**Rating:** 3
**Confidence:** 3

**Summary:**

The authors studies simple of neural encoding. The question is whether two
distinct goals, accurate transmission of information and learning the
distribution of environmental stimuli can be achieved simultaneously. The
authors argue that yes, it can, using the key assumption of a uniformly
partitioned input space. The coding principle of the authors is finally applied
to image patches, where it yields edge-like features.

**Strengths:**

- The author studies an important question, namely simple coding schemes that
  reproduce filters that resemble those of deep convolutional networks or parts
  of the visual system.
- The author develops intuitions in simple toy models before moving to
  applications on real images.
- The filters shown in Fig. 3 bear a striking resemblance to the filters of a
  trained VGG model (although I have some questions on the methodology, see
  below)



**Weaknesses:**

I found the article confusing to read in a few places. For example, early on,
the author states that "maximising the rate of transmission" is equivalent to
maximising the entropy of the output distribution $H_Q$. I would think that what
you transmission of information requires maximising the mutual information $I(X;
Q)$ between the distribution over inputs and outputs. (around eqs 1 + 2; note
that the notation is rather confusing here, using lower-case $p$ for the
distribution over input stimuli $x$, and capital $Q$ for the distribution over
output states $y$). Why are you maximising simply the entropy of the
distribution over outputs?

Similarly, in the section on the even code principle, I'm confused by the
question of how the IPU models the input distribution. The way I read Sec. 2,
the IPU is considered a function of the stimuli $y=f(x)$ -- in that sense, it
doesn't model the input distribution, we cannot sample from it. It can give a
more or less faithful representation of $x$, as measured for example by mutual
information if the mapping is probabilistic,

As you then move on to learn two pixel distributions, I'm confused about your
use of MLPs. MLPs are powerful neural networks, but you seem to use them to
"learn" to partition the input space into equal partitions - is this not
possible by just writing down a simpler model?

Given my trouble understanding the first few sections, I cannot competently
comment on the experimental results - while the filters obtained by the authors
do bear a striking resemblance to the filters of a VGG network, I don't really
understand how the author obtained them. Some additional clarifications would
therefore be more than welcome.


**Questions:**

See above

**Limitations:**

See above

---

> ### Author Rebuttal · Authors · 2023-08-05
>
> Thank you for your constructive feedback. We will address your concerns and questions point by point.
>
> * Regarding maximizing the entropy of the output distribution
>
> In Appendix A, we have proven that for our model, maximizing mutual information (rate of transmission), defined as $H_Q - H(y|x)$, is equivalent to maximizing the entropy of the output distribution.
>
> * Regarding the use of capital $Q$
>
> The seemingly unconventional notation is purposeful. In this paper, we use a capital letter to denote distribution over the output states $y$, and a lower-case letter for distributions over the input states $x$. In Section 3, we introduced the quantity $q(x)$, which is the translation of $Q(y)$ into the input space. Using $Q(y)$, instead of the seemingly natural lower-case notation, clarifies that it is a different function from $q(x)$. We will explain the notation in the text to prevent confusion.
>
> * Regarding how the IPU models the input distribution
>
> The function that IPU learns, $y=f(x)$, is a deterministic step function that evenly partitions the input probability distribution. Each value of y maps to an equal portion of $p(x)$. Consequently, we can translate the output probability distribution $Q(y)$ (a constant) into an probability distribution learned by the model $q(x)$ in the input space using Eq. (3) and (4). $q(x)$ is a discretized approximation of the real probability distribution $p(x)$ (see Fig. 2 in the attached rebuttal pdf).
>
> * Regarding the use of MLPs to model two-pixel distribution
>
> The MLP is used to approximate the function $y=f(x)$. The partitions were visualized using the `tricontour` function from `matplotlib`. Using MLPs to create a set of equally spaced parallel lines, as in Fig. 1(a), might seem like overkill. However, an information processing system should be general and work with any 2D (or higher-dimensional) distributions (see Fig. 1 of the one-page rebuttal pdf for an example). The partitioning must be learned by the IPU, which initially has no knowledge about the data it will encounter. For this reason, we use a versatile and powerful function approximator like MLP. Another reason is that we want to use the same kind of model for all numerical studies, and MLP serves this goal very well.
>
> * Regarding Fig. 3
>
> This figure does not compare the filters learned by the model but studies how well the model embeds image patches in its representation space according to image similarity viewed by the model. The generation of this figure is described in its caption and the subsection "5.2.2 Image Patch Similarity" To add more detail, we used the following steps:
> 1. We randomly sampled 1 million 5x5 color image patches from the datasets.
> 2. For each image patch, we used our model to generate a representation, which is a binary vector of size 96 (12 bytes).
> 3. For each image patch, we used the first 10 layers of a pretrained VGG16 model (the `torchvision` default VGG16 model) to generate a representation, which is a float vector of size 128 (512 bytes).
> 4. We selected 16 random image patches out of the 1 million and plotted them as the first column in Fig. 3 (a) and (b).
> 5. For each of the 16 random images, we calculated the distance to the remaining 999,999 images using the representations we got in steps 2 and 3. Then, we chose the top 9 image patches with the smallest distances and showed them in the remaining 9 columns in Fig. 3 (a) and (b) in order, respectively.
> Our simple model's representation produces comparable results to those generated with VGG16 while using less than 3% of the storage space. This illustrates the efficiency of our model.
>
> I hope the above explanations answer your questions and make our paper more understandable. Should you have any further questions, we would be glad to clarify them. Thank you for the time and effort you have put into evaluating our paper.

---

> > ### Comment · Reviewer_aZBn · 2023-08-14
> > **Thank you for your reply**
> >
> > I have read your reply. I appreciate the clarifications on my question; I think those points should be clarified in an eventual revision. Looking through the other reviews and the respective rebuttals, I think that this process has unearthed a few directions in which the paper could be strengthened. In the meantime, I will keep my score.

---

### Author Rebuttal · Authors · 2023-08-05

Attached are the two figures referenced in the rebuttal.

---

### Decision · Program_Chairs · 2023-09-21

**Decision:**

Reject

**Comment:**

Reviewers find the work novel, but also found it rather abstract  and believe the paper needs more contextualization and linking to the biology.   There are several directions suggested by the reviewers that will hopefully make this paper stronger for subsequent submission.